# Design and Optimization of a Five-Phase Permanent Magnet Synchronous Machine Exploiting the Fundamental and Third Harmonic

Mouna Oukrid, Nicolas Bernard *, Mohamed-Fouad Benkhoris and Djamel Ziane

Nantes Atlantique Electrical Energy Research Institute (IREENA), University of Nantes, UR 4642, F-44600 Saint-Nazaire, France; mouna.oukrid@univ-nantes.fr (M.O.); mohamed-fouad.benkhoris@univ-nantes.fr (M.-F.B.); djamel.ziane@univ-nantes.fr (D.Z.)
* Correspondence: nicolas.bernard@univ-nantes.fr

**Abstract:** This paper deals with the design of five-phase permanent magnet synchronous machines (PMSMs) exploiting the third harmonic for torque generation. Through the optimization of the stator size and rotor structure, the objective functions related to mass and electric losses are minimized for a targeted electromagnetic power (10 kW and 400 rpm) and a given volume. The study takes into account saturation, thermal, electrical and mechanical constraints. On that note, a 1D analytical magnetic model, considering the existence and use of the third harmonic, is presented. The design optimization then shows how the use of harmonic 3 can improve the machine's performance. It will be shown that, for a given electromagnetic torque, taking the third harmonic into account in the sizing process leads to a mass reduction that can reach 20% and electrical losses that can go up to 21%. A finite element analysis model of the five-phase PMSM is then established in order to verify the results of the optimization and validate them.

**Keywords:** PMSG; five-phase machines; secondary machine; optimization; harmonics

## 1. Introduction

In the current context, rotating machines are becoming increasingly important due to the continuous growth in electric mobility and renewable energies. Therefore, the need for more efficient, higher performance electrical machines is increasing [1]. As a result, research into multiphase machines has grown due to their multiple advantages, particularly their potentially high power density and high fault-tolerance capability. In a five-phase machine, for example, it is possible to exploit the third harmonic to generate a torque, which provides an additional possibility to boost performance. In addition, it allows for the use of limited voltage and current power converters in high-power applications [2–4]. In the scientific literature, studies dealing with the design and control of multiphase machines mainly focus on high-power applications for different fields, such as marine current turbines [5], offshore wind turbines [6], and aerospace applications [7]. In these applications, where the minimization of mass in a reduced volume is the main objective, the use of a secondary machine is an interesting solution [8]. In [5], the authors show, for a marine turbine application, the advantages of a five-phase machine with the exploitation of the third harmonic compared to a three-phase machine in terms of energy conversion quality (less torque ripple) in normal conditions and under open-circuit fault conditions. In the same way, in [9], it is demonstrated that a five-phase machine can provide a higher torque (about 15%) and less pulsating torque (71% lower) compared to a three-phase machine with the same copper losses. In [10], an optimal third-harmonic injection strategy is proposed with the goal of minimizing copper loss at a given torque. Other papers focused on the implementation of a control strategy [11] either in a normal operating mode [12] or in post-fault operation [13]. Predictive control for multiphase machines was also discussed

in [14]. In [15], the ratio of torque to copper loss is increased with an online identification. In [8,16], a third-harmonic current injection is also presented and applied to a double-polarity five-phase machine in which the torque can be generated by only the first, third, or both first and third sinusoidal currents. In these papers, some design considerations are presented, such as the choice of slot/pole combination and the rotor structure.

However, the design optimization of a PMSM requires an analysis of the parameters of the machine while considering multiphysics constraints. Then, following an analysis similar to that in [17], the aim of this paper is to present a design process integrating a thermal model and considering not only copper losses, as is usually done, but also iron losses. This study allows a comparison to be made between the optimized PMSM using the fundamental harmonic and the optimized PMSM using both the fundamental and third harmonics.

In the case of high-power machines, for example, with concentrated windings made of one slot per pole and per phase, the existence of a third harmonic on the electromotive force can be used to produce additional torque without increasing voltage and magnetic constraints, as will be shown further.

This paper aims to present an optimal design that allows the mass of the machine to be minimized as well as the losses produced for a targeted electromagnetic power by using a multi-objective algorithm.

The paper is organized as follows. Section 2 introduces the analytical model of the PMSM used, considering both the fundamental and third harmonics. In Section 3, the optimization problem with the optimization parameters, objectives, and constraints is presented. The results of the optimization are given and compared with an optimization considering only the first harmonic. In this part, we analyze the effect of the presence of the third harmonic. In Section 4, a finite element analysis is carried out in order to validate the design of the optimum selected machine.

## 2. Analytical Modeling

This section presents a 1D analytical model of the five-phase PMSM which will be used in the optimization process. Figure 1 shows the design and the main geometrical parameters of a surface-mounted permanent magnet machine. The inner reduced radius of the stator, $r_s$, and rotor, $r_o$, as well as the winding's outer reduced radius, $r_w$, are important parameters that will affect the design of the PMSM. The different widths of the rotor yoke, $W_{ry}$, stator yoke, $W_{sy}$, and permanent magnet, $W_{PM}$, as well as the airgap thickness, $W_{ag}$, are also studied, as their effect on the machine's performance is also noticeable.

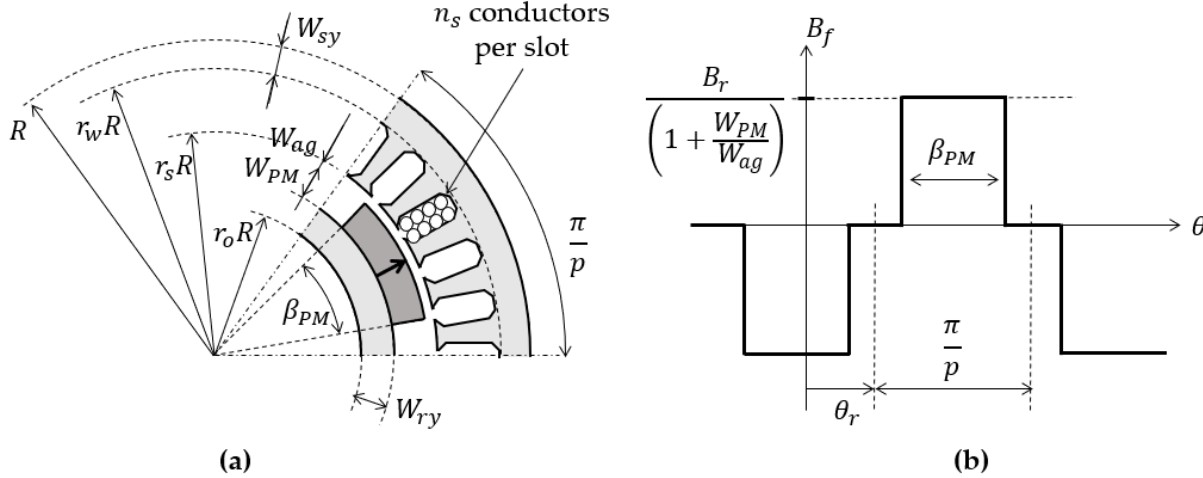

**Figure 1.** Design and geometric parameters of the PMSM (**a**); airgap flux density of open circuit (**b**).

A polyphase system could be decomposed into several orthonormal single-phase or two-phase systems that are mechanically coupled and magnetically independent. Thus, each system, which can also be called a fictitious machine, can be managed independently [18,19].

While three-phase machines can be controlled in a single subspace obtained by implementing a Concordia transformation $(\alpha; \beta)$, a five-phase machine can be controlled using two orthogonal subspaces defined by a Concordia transformation. Each subspace represents a fictitious machine. The principal subspace $(\alpha_p; \beta_p)$ has electromotive forces $\left(E_{\alpha_p}, E_{\beta_p}\right)$ and a cyclic inductance denoted $L_p$. The secondary subspace $(\alpha_s; \beta_s)$ has electromotive forces $(E_{\alpha_s}, E_{\beta_s})$ and a cyclic inductance denoted $L_s$. This decomposition leads to one single-phase system called the homopolar and two two-phase systems, as shown in Figure 2.

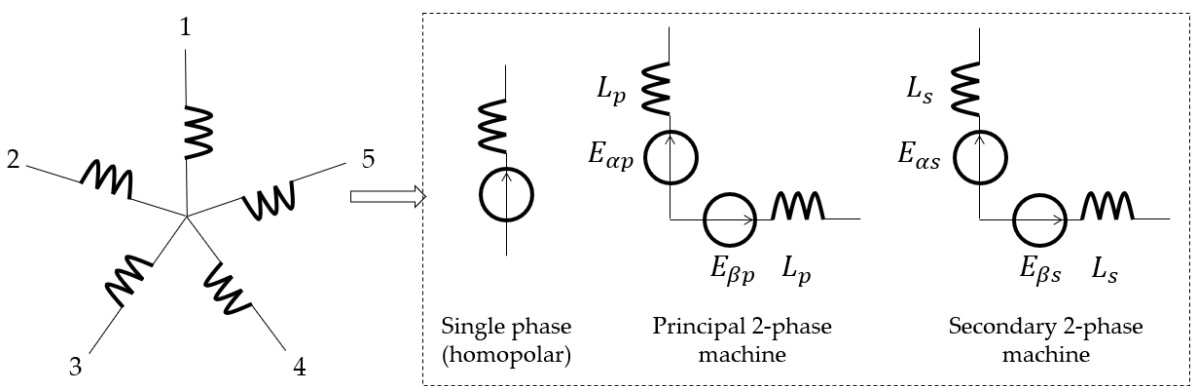

**Figure 2.** Five-phase machine decomposition.

In the case of a sinusoidal electromotive force, only one of the three fictitious machines exist and can produce torque. The two-phase machine which is linked to the harmonic of rank 1 is called the main machine, and the two-phase machine linked to the third harmonic is called the secondary machine. As a result, the second two-phase machine will be able to produce torque mainly thanks to rank 3 harmonics [20,21]. The concept of fictious machines and the decomposition of multiphase machines are especially useful for the study of control.

### 2.1. Magnetic Model of the PMSM

In this paper, it is assumed that the ferromagnetic parts are linear and of infinite permeability. It is also assumed that the thickness of the laminations is sufficiently thin to consider the iron reluctance negligible at the third harmonic. As already demonstrated [10], in the case of a maximum torque per RMS current control (MPTA), the third-harmonic current is optimum when it is in phase with the third-harmonic voltage. Then, we assume that $i_{d1} = 0$ and $i_{d3} = 0$.

#### 2.1.1. Flux Densities

The airgap flux density created by the magnets is represented in Figure 1b. Under each pole, its magnitude is obtained from Ampere's law [22]. It can be written as a Fourier series, where each harmonic component, $B_{hf}(\theta, \theta_r)$, of odd rank, $h$, is expressed as follows:

$$B_{hf}(\theta, \theta_r) = \frac{4}{h\pi} \frac{B_r}{\left(1 + \frac{W_{PM}}{W_{ag}}\right)} sin\left(\frac{hp\beta_{PM}}{2}\right) sin(hp(\theta - \theta_r)) \tag{1}$$

where $B_r$, $\beta_{PM}$, $W_{PM}$, and $W_{ag}$ are, respectively, the remanent flux density, the arc of the magnets, the width of the magnets, and the airgap thickness.

The airgap flux density created by the stator can be expressed by the magnetomotive force, *Fmm*, and the surface permeance, $\mathcal{P}$. For each odd harmonic, the harmonic airgap flux density is as follows:

$$B_{hs}(t,\theta) = Fmm_h(t,\theta)\mathcal{P} \tag{2}$$

where $Fmm_h(t,\theta)$ is the distribution of the harmonic magnetomotive force produced by the five phases. With $n_s$, the number of conductors per pole and per phase, and an MPTA control, the magnetomotive force can be expressed as follows:

$$Fmm_h(t,\theta) = \frac{5}{h\pi}n_s I_{sh}\cos(h(\omega t - p\theta - \psi_h)) \tag{3}$$

where $I_{sh}$ is the current harmonic of rank $h$ and $\psi_h$ is its phase angle.

If slot effects are neglected, the surface permeance is constant and given as follows:

$$\mathcal{P} = \frac{\mu_0}{(W_{ag} + W_{PM})} \tag{4}$$

Thanks to Gauss's law, it is possible to express the magnitude of the flux densities in the stator tooth ($B_{stm}$), stator yoke ($B_{sym}$), and rotor yoke ($B_{rym}$) as a function of the magnitude of the resultant airgap flux density ($B_{agm}$). They are, respectively, given by the following:

$$\begin{cases} B_{sym} = \frac{r_s B_{agm}}{p(1-r_w)} \\ B_{stm} = \frac{B_{agm}}{k_t} \\ B_{rym} = \frac{r_s B_{agm}}{p\left(\frac{r_s R - W_{ag} - W_{PM}}{R} - r_o\right)} \end{cases} \tag{5}$$

where $k_t$ is the tooth-opening-to-slot-pitch ratio.

When the problem is limited to the first harmonic ($h = 1$) and a control with $i_{d1} = 0$, the magnitude of the resultant airgap flux density can be analytically written as follows:

$$B_{agm} = B_{1agm} = \sqrt{B_{1fm}^2 + B_{1sm}^2} \tag{6}$$

On the contrary, for a study considering the third harmonic ($h = 1$ and $h = 3$), the magnitude of the resultant airgap flux density cannot be easily expressed analytically.

$$B_{agm} = max\left(\sum_{h=1,3} B_{hf}(\theta) + \sum_{h=1,3} B_{hs}(\theta)\right) \tag{7}$$

2.1.2. Torque

The electromagnetic power of the PMSG can be expressed using the induced e.m.f of each phase of the machine and the currents of each phase for the fundamental and third harmonics:

$$P_{emh} = \sum_{k\epsilon\{1,2...5\}} E_{hsk}I_{hsk} = 5p\Omega B_{hfm}n_s I_{hs}R_s L \tag{8}$$

where $\Omega$ is the mechanical angular velocity, $R_s$ is the stator's inner radius, and $L$ is the active length of the machine.

We can deduce the electromagnetic torque of the machine:

$$\Gamma_{em_h} = \frac{P_{emh}}{\Omega} = 5p B_{hfm}n_s I_{hs}R_s L \tag{9}$$

*2.2. Loss Model*

In this part, we consider the copper losses as well as the iron losses, taking into account the first and third harmonics.

2.2.1. Copper Losses

Copper losses depend on the RMS current, as follows:

$$P_c = 5 \, R_c \, I_{RMS}^2 \tag{10}$$

where the rms stator current is as follows:

$$I_{RMS} = \sqrt{I_{1s}^2 + I_{3s}^2} \tag{11}$$

An expression for the copper resistance, considering a slot fill factor $k_f$ and the correcting length coefficient $k_L$, can be written:

$$R_c = \frac{2p \, k_L L n_s^2}{\sigma_c \, k_f \, S_s} \tag{12}$$

with the slot cross-section given as follows:

$$S_s = \frac{\pi R^2 \left(r_w^2 - r_s^2\right)(1 - k_t)}{Z_s} \tag{13}$$

Then, for a five-phase winding with one slot/pole/phase ($Z_s = 10p$), this gives:

$$R_c = \frac{20}{\pi} \frac{k_L L \, n_s^2}{\sigma_c \, k_f \, (1 - k_t)} \frac{p^2}{R^2 (r_w^2 - r_s^2)} \tag{14}$$

If we consider the end-winding separately, we can express the end-winding losses as follows:

$$P_{ew} = 5 \, R_{ew} \, I_{RMS}^2 \tag{15}$$

with the expression of the end-winding resistance given as follows:

$$R_{ew} = \frac{20}{\pi} \frac{L_{ew} \, n_s^2}{\sigma_c \, k_f \, (1 - k_t)} \frac{p^2}{R^2 (r_w^2 - r_s^2)} \tag{16}$$

and the end-winding length expressed using the end-winding coefficient, $k_L$ :

$$L_{ew} = (k_L - 1)L \tag{17}$$

2.2.2. Iron Losses

Several studies have been carried out on the estimation of iron losses [23,24]. We apply here the principle of the separation of losses, including both hysteresis losses and eddy current losses. Additional losses due to magnetic anomalies, metallurgical processes, and rotating fields were considered by introducing an additional coefficient, $k_{ad}$. Neglecting iron losses in the rotor and considering the contribution of the first and third harmonics, the losses in the stator teeth ($P_{mgt}$) and stator yoke ($P_{mgy}$) are, respectively, written as follows:

$$P_{mgt} = \sum_{h=1,3} k_{ad} \left( K_H p h (2\pi f) + K_F p^2 h^2 (2\pi f)^2 \right) M_{st} B_{hstm}^2 \tag{18}$$

$$P_{mgy} = \sum_{h=1,3} k_{ad} \left( K_H p h (2\pi f) + K_F p^2 h^2 (2\pi f)^2 \right) M_{sy} B_{hsym}^2 \tag{19}$$

The sum, $P_{tot}$, of the electrical losses presented in the equations above (Equations (10), (18) and (19)) defines the first objective (mass being the second one):

$$P_{tot} = P_c + P_{mgt} + P_{mgy} \tag{20}$$

*2.3. Constraints*

2.3.1. Mechanical Constraints

The generator's geometric and mechanical constraints must allow it to be mechanically feasible. On the other hand, the thickness of the stator yoke and teeth has a major influence on machine deformation and noise [25]. To avoid excessive stress, these parameters were limited to a minimum width, noted $W_{min}$, of 20 mm.

The skin effect should also be considered as a constraint on the thickness of the lamination, $th_{lam}$, particularly for the third harmonic, where the skin depth can be estimated using the iron conductivity, $\sigma_{iron}$, and permeability as follows:

$$\delta = \sqrt{\frac{1}{3\pi f \mu_0 \mu_r \sigma_{iron}}} \tag{21}$$

Thus, we consider here the following constraints:

$$\begin{cases} r_s < r_w \\ r_o < r_s \\ W_{sy} > W_{min} \\ W_{ry} > W_{min} \\ W_{ag} > W_{ag\ min} \\ th_{lam} \geq 2\delta \end{cases} \tag{22}$$

2.3.2. Electrical Constraints

We assume here a two-level back-to-back voltage source converter. In this case, the voltage limit must not exceed half the DC bus voltage, $U_{DC}$ [22,26].

This maximum voltage is obtained from the sum of the harmonic voltages $V_{1s}$ and $V_{3S}$. If the harmonic currents are in phase with the back e.m.f voltages, they are expressed as follows:

$$V_{1s}(t) = \sqrt{(E_{1s} + R_c I_{1s})^2 + (L_p \omega I_{1s})^2} \, sin\left(\omega t + atan\left(\frac{L_p \omega I_{1s}}{E_{1s} + R_c I_{1s}}\right)\right) \tag{23}$$

$$V_{3s}(t) = \sqrt{(E_{3s} + R_c I_{3s})^2 + (3L_s \omega I_{3s})^2} \, sin\left(3\omega t + atan\left(\frac{3L_s \omega I_{3s}}{E_{3s} + R_c I_{3s}}\right)\right) \tag{24}$$

with the cyclic inductances expressed as a function of the self-inductance, $L_0$, as follows:

$$\begin{cases} L_p = \frac{5}{2} L_0 \\ L_s = \frac{5}{2} \frac{L_0}{9} \\ L_0 = \frac{4\mu_0 n_s^2 R r_s L}{\pi W_{PM}} \end{cases} \tag{25}$$

Assuming linear conditions, we can write

$$max(V_{1s}(t) + V_{3s}(t)) \leq \frac{U_{DC}}{2} \tag{26}$$

For the voltage limit and considering the targeted power (10 kW), the choice was made to set the DC bus voltage at 600 V.

2.3.3. Saturation Constraints

The magnitude of the flux density must be limited in each magnetic part of the machine [17,22]. The flux density in the stator yoke ($B_{sym}$), the stator teeth ($B_{stm}$), and the rotor yoke ($B_{rym}$) is limited to $B_{sat}$ with the choice of $B_{sat} = 1.6\ T$.

$$\begin{cases} B_{sym} < B_{sat} \\ B_{stm} < B_{sat} \\ B_{rym} < B_{sat} \end{cases} \qquad (27)$$

### 2.3.4. Thermal Constraints

Thermal analysis is mandatory during the design process of an electric machine, especially in applications in which the machine is expected to operate at its temperature limits. Advances in computers over the last few years have resulted in powerful tools for the thermal analysis of electric machines. The available methods can be grouped into two major categories: analytical lumped-parameter thermal networks (LPTNs) and numerical methods. In our case, in order to reduce the computation time required for the optimization process, the use of a lumped-parameter thermal model is adaptable, as shown in [27–29]. In fact, this method allows us to assign to each node and component a parameter which is located in the system where the heat transfer happens. The lines represent the path where the heat can flow, and the arrows present the direction of this transfer.

Figure 3c represents the proposed thermal model with the thermal resistances and losses considered for the calculation. Due to the symmetry, it is sufficient to model half a slot pitch. We assume there is no heat exchange with the air gap and a homogeneous temperature in the end-windings. Each cylindrical part of the stator, as represented in Figure 3a, is modeled by an equivalent circuit (see Figure 3b) made up of a heat source ($P$), two thermal resistances ($\mathcal{R}_{r1}$ and $\mathcal{R}_{r2}$) for radial heat transfer, and two equal thermal resistances for orthoradial heat transfer ($\mathcal{R}_t$). For the axial conduction heat transfer, the resistance between the winding and the end-winding is considered only (represented in blue in Figure 3c).

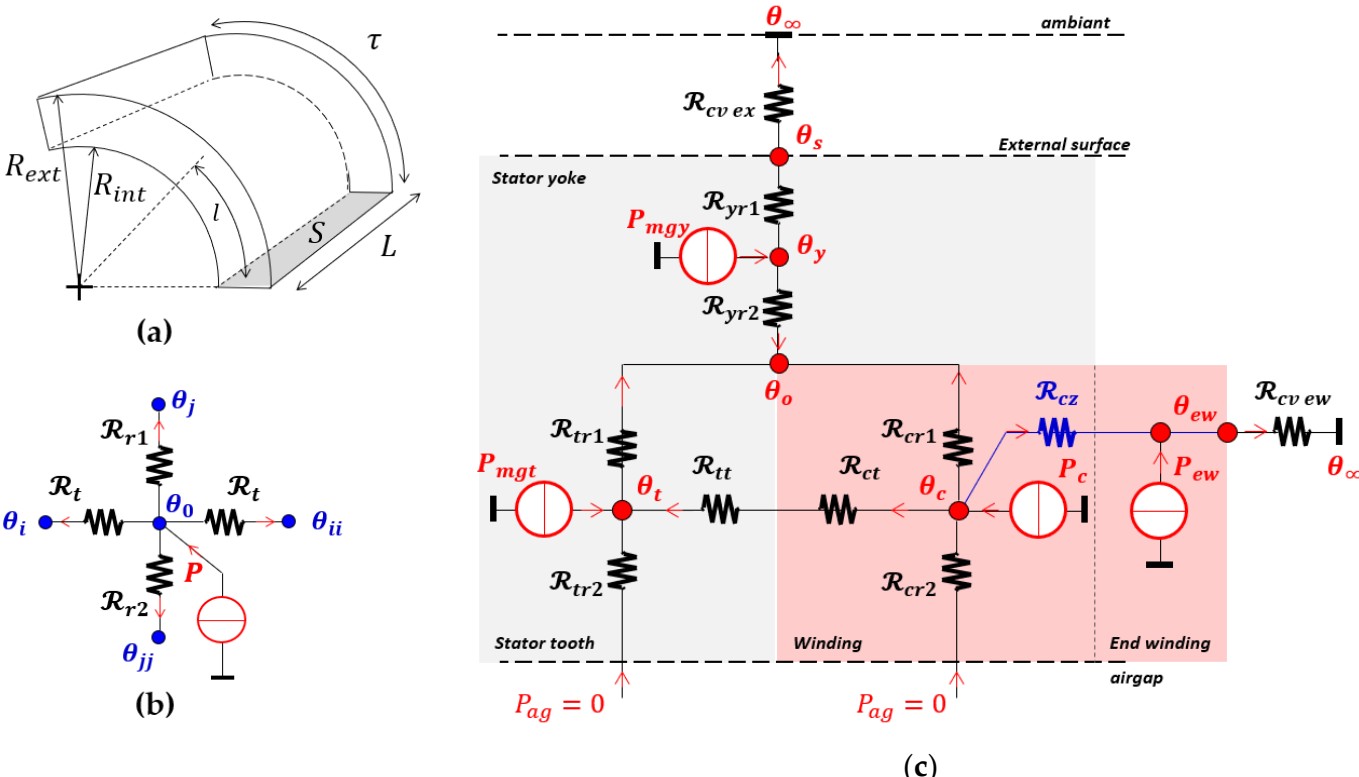

**Figure 3.** Cylindrical piece (**a**) with its equivalent thermal model (**b**) and the thermal model of the machine (**c**).

In the radial direction, we can calculate the expression of resistances $\mathcal{R}_{r1}$ and $\mathcal{R}_{r2}$ according to the geometric parameters shown in Figure 3a, as follows:

$$R_{r1} = \frac{1}{2\lambda_{mat}\tau L}\left[\frac{2\left(\frac{R_{ext}}{R_{int}}\right)^2 ln\left(\frac{R_{ext}}{R_{int}}\right)}{\left(\frac{R_{ext}}{R_{int}}\right)^2 - 1} - 1\right] \tag{28}$$

$$R_{r2} = \frac{1}{2\lambda_{mat}\tau L}\left[1 - \frac{2ln\left(\frac{R_{ext}}{R_{int}}\right)}{\left(\frac{R_{ext}}{R_{int}}\right)^2 - 1}\right] \tag{29}$$

where $\lambda_{mat}$ is the material's conductivity.

In the orthoradial direction, both resistances $\mathcal{R}_t$ have a constant rectangular cross-section, $S$. They are equal and calculated as follows:

$$\mathcal{R}_t = \frac{1}{\lambda_{mat}}\frac{l}{S} \tag{30}$$

with

$$l = \frac{R_{ext} + R_{int}}{2}\frac{\tau}{2} \tag{31}$$

and

$$S = (R_{ext} - R_{int})L \tag{32}$$

The resistance between the winding and the end-winding in the axial direction, depending on the slot cross-section, $S_s$, is expressed as follows:

$$\mathcal{R}_z = \frac{L}{\lambda_{mat}S_s} \tag{33}$$

For the convection heat transfer, the thermal resistance can be calculated using the heat transfer coefficient, $h$, with $h = 100\ W/m^2K$ for air-cooled convection [22]. For the radial stator frame, considering half a slot pitch, we write the following:

$$R_{cv\ ex} = \frac{1}{h\left(\frac{\pi RL}{Z_s}\right)} \tag{34}$$

and for the end-windings:

$$R_{cv} = \frac{1}{h\ S_{ew}} \tag{35}$$

with an area $S_{ew}$ that can be expressed as follows [30]:

$$S_{ew} = \frac{\pi}{2Z_s}R_w L + \frac{\pi\left(R_w^2 - R_s^2\right)}{4Z_s} \tag{36}$$

Finally, using the equivalent circuit and the calculated thermal resistances, we can calculate the temperature at each node. At each node of an elementary pattern of the LPTN (Figure 3b), the following equation can be written:

$$\frac{\theta_0 - \theta_i}{\mathcal{R}_t} + \frac{\theta_0 - \theta_{ii}}{\mathcal{R}_t} + \frac{\theta_0 - \theta_j}{\mathcal{R}_{r1}} + \frac{\theta_0 - \theta_{jj}}{\mathcal{R}_{r2}} = P \tag{37}$$

The resulting system of equations can be presented in matrix form:

$$[U] = [\theta][A] \tag{38}$$

where $[A]$ is the thermal conductivity matrix. $[U]$ and $[\theta]$ are, respectively, the vector of losses and the vector of temperatures, written as follows:

$$[U]^t = \left[ \frac{\theta_\infty}{R_{cv_{ex}}} \; ; P_{mgy} \; ; 0 \; ; P_{mgt} \; ; \; P_c \; ; P_{ew} \right] \tag{39}$$

$$\left[ \theta \right]^t = \left[ \theta_s \; ; \theta_y \; ; \theta_o \; ; \theta_t \; ; \theta_c \; ; \theta_{ew} \right] \tag{40}$$

The temperature at each node of the equivalent thermal model is obtained since Equation (37), and can be written as follows:

$$[\theta] = [A]^{-1}[U] \tag{41}$$

Mass calculation:

Only the masses of the active parts will be considered here. $M_c$, $M_{st}$, $M_{sy}$, $M_{ry}$, and $M_{PM}$ are, respectively, the mass of the copper, stator teeth, stator yoke, rotor yoke, and magnets. They are calculated as functions of the geometric parameters of the machine and the different densities of each material, $\rho$, as shown below:

$$M_c = \pi \left( r_w^2 - r_s^2 \right) k_r R^2 L \rho_c \tag{42}$$

$$M_{st} = \pi k_t \left( r_w^2 - r_s^2 \right) R^2 L \rho_{Iron} \tag{43}$$

$$M_{sy} = \pi \left( 1 - r_w^2 \right) R^2 L \rho_{Iron} \tag{44}$$

$$M_{ry} = \pi \left( \left( r_s R - W_{mag} \right)^2 - R^2 r_o^2 \right) L \rho_{Iron} \tag{45}$$

$$M_{PM} = 2\pi R \left( r_s R - W_{ag} - \frac{W_{PM}}{2} \right) \left( \frac{L}{R} \right) W_{PM} \beta_{PM} \rho_{PM} \tag{46}$$

$$Mass = M_c + M_{st} + M_{sy} + M_{ry} + M_{PM} \tag{47}$$

## 3. Optimization

The machine is sized using a genetic optimization algorithm (NSGA II [31]). The problem proposed here is to minimize mass and losses for a given electromagnetic power in a given volume. We will consider the following two cases: optimization without the exploitation of the third harmonic and optimization with the exploitation of the third harmonic. In order to compare these two optimizations and quantify the advantage of exploiting the third harmonic, we introduce the quantity $\gamma$, the ratio between the rms currents $I_{3s}$ and $I_{1s}$, as follows:

$$I_{3s} = \frac{\gamma}{\sqrt{1+\gamma^2}} I_s I_{1s} = \frac{1}{\sqrt{1+\gamma^2}} I_s I_s^2 = I_{1s}^2 + I_{3s}^2 \tag{48}$$

Besides the thermal, mechanical, and saturation constraints defined by the properties of the machine's material, a voltage limit, $V_{limit}$, (which is equal to half the DC bus voltage) is also imposed by the power electronics converter. Here, the electromagnetic power and the speed are set, respectively, at 10 kW and 400 rpm, within a volume limit determined by $R_{ext\;max} = 19$ cm and $L_{max} = 9.5$ cm. This choice was led by the realization of a prototype, which will be used in our future studies. This machine (see Figure 4) is made of six pole pairs and 180 slots. It can operate either as a fifteen-phase PMSM with one slot/pole/phase or as a five-phase PMSM with three slots/pole/phases. In future work and in a further paper, this machine will be studied in the five-phase configuration.

**Figure 4.** Prototype: (**a**) stator, (**b**) winding assembly, and (**c**) stator with its winding.

Thus, the problem statement is as follows:
Objectives:

$$\min_{x}\left(P_{tot} = P_c + P_{mgy} + P_{mgt}\right) \min_{x}\left(Mass = M_c + M_{st} + M_{sy} + M_{ry} + M_{PM}\right) \tag{49}$$

Constraints:

$$\begin{aligned}
max\{\theta_{ws}(t); \theta_{wr}(t); \theta_{wer}(t); \theta_{wer}(t)\} &\leq \theta_{max} = 145\,°\text{C} \\
max\{B_{stm}(t); B_{sym}(t); B_{rtm}(t); B_{rym}(t)\} &\leq B_{sat} = 1.6\,\text{T} \\
max\{V_{max}(t)\} &\leq V_{limit} = 300\,\text{V} \\
min\{W_{sy}; W_{ry}\} &\geq W_{min} = 20\,\text{mm} \\
min\{th_{lam}\} &\geq 2\delta
\end{aligned} \tag{50}$$

with the following optimization decision variables:

$$x = \left(p, R, r_s, r_w, r_0, W_{mag}, B_r, \gamma, \beta_{PM}\right)^T \tag{51}$$

and $W_{mag}$, the magnetic airgap, defined as follows:

$$W_{mag} = W_{PM} + W_{ag} \tag{52}$$

To achieve this bi-objective optimization, the multi-objective genetic algorithm Non-dominated Sorting Genetic Algorithm II (NSGA-II), developed by [31], was used. This algorithm is well known today and is very often used for its good performance and ease of use. The algorithm and the method for optimizing the two objectives are described in reference [28]; notably, the selection process was carried out by a crowded comparison operator, which led to a uniformly spread-out Pareto optimal front.

*3.1. Constant Parameters*

The constant parameters used for the optimization are summarized in Table 1.

**Table 1.** Constant parameters.

| Parameters | Values |
| --- | --- |
| $W_{PM}/W_a$ | 3/7 |
| $k_t$ | 0.5 |
| $L_{max}$ | 9.5 cm |
| $R_{max}$ | 19 cm |
| $N$ | 400 tr/min |
| $k_f$ | 0.5 |
| $\rho_c$ | 8900 kg/m$^3$ |
| $\rho_{Iron}$ | 7800 kg/m$^3$ |
| $\rho_{PM}$ | 7400 kg/m$^3$ |
| $\sigma_c$ | $59 \times 10^6$ S/m |
| $k_L$ | 1.2 |

**Table 1.** *Cont.*

| Parameters | Values |
| --- | --- |
| $k_{ad}$ | 2 |
| $K_H$ | 0.0019 Wm$^3$/kg$^2$T$^2$Hz$^2$ |
| $K_F$ | $8.33 \times 10^{-7}$ Wm$^3$/kg$^2$T$^2$Hz$^2$ |
| $\lambda_c$ | 5 W/m K |
| $\lambda_{Iron}$ | 25 W/m K |
| $\theta_\infty$ | 25 °C |
| $h$ | 100 W/m$^2$K |

### 3.2. Results

From NSGA-II, we collected the different results of the optimization problem. We used a MATLAB code [32] with different numbers of generations and population sizes going up to 4000 (see Table 2). These parameters, for the problem considered here (where the results are presented in Figure 5), lead to a reduced computation time (a few minutes per optimization) while enabling good convergence and robustness of the results (see Figure 6).

**Table 2.** Main parameters of NSGA II.

| Parameters | Value |
| --- | --- |
| Population size | 4000 |
| Number of generations | 4000 |
| Distribution index for crossover | 20 |
| Distribution index for mutation | 50 |

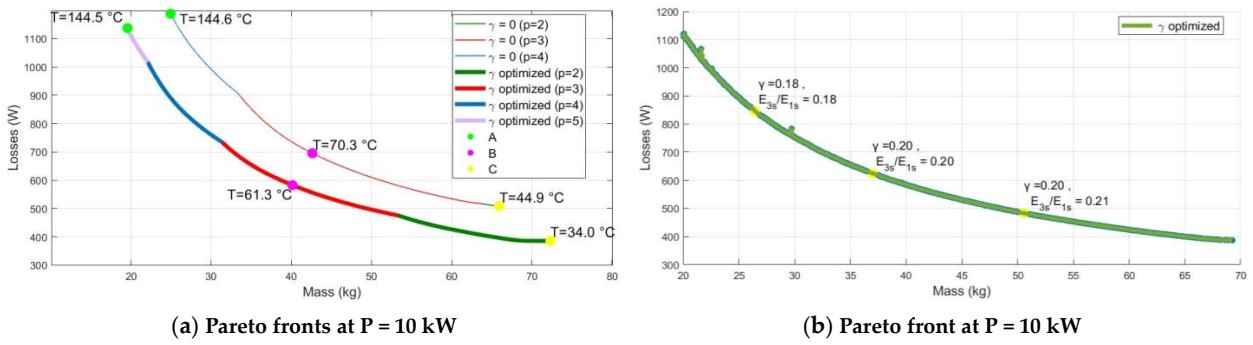

| (**a**) Pareto fronts at P = 10 kW | (**b**) Pareto front at P = 10 kW |
| --- | --- |

**Figure 5.** Pareto optimal fronts with γ = 0 (**a**) and optimized γ; visualization of optimized γ and the ratio E$_{3s}$/E$_{1s}$ (**b**).

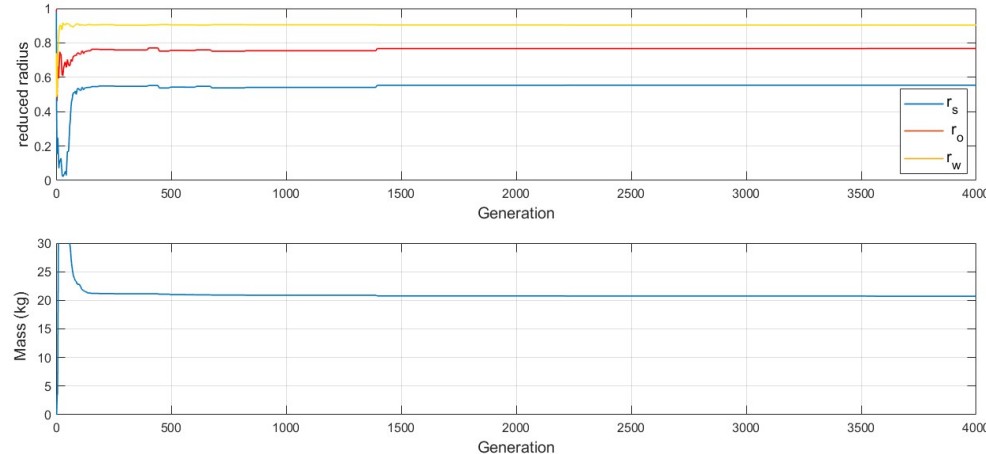

**Figure 6.** Convergence of reduced radii (**top**) and the mass objective (**bottom**) for γ optimized and machine A.

In order to illustrate the benefits of exploiting the third harmonic in a concentrated winding machine, an optimization was carried out for both cases: $\gamma = 0$ and $\gamma$ optimized. The Pareto fronts obtained, represented in Figure 5a, clearly show the benefits of exploiting the third harmonic. We can also observe that the optimized ratio, $\gamma_{opt}$, leads to the third-harmonic current being about 20% of the first-harmonic current (see Figure 5b), which is equal to the ratio, $E_{3s}/E_{1s}$, between the third-harmonic e.m.f and the first-harmonic e.m.f.

For this study, three optimal machines are pointed out at the Pareto optimal front, marked with points A, B, and C. Machines 'A' and 'C' at the extreme point of the front represent, respectively, the optimum result considering the mass criterion only and the optimum result considering the loss criterion only. Machine 'C' represents the result considering a combination of the two objectives. In order to analyze the effect of the decision variables, their evolution for each element of the population is shown in Figures 7 and 8.

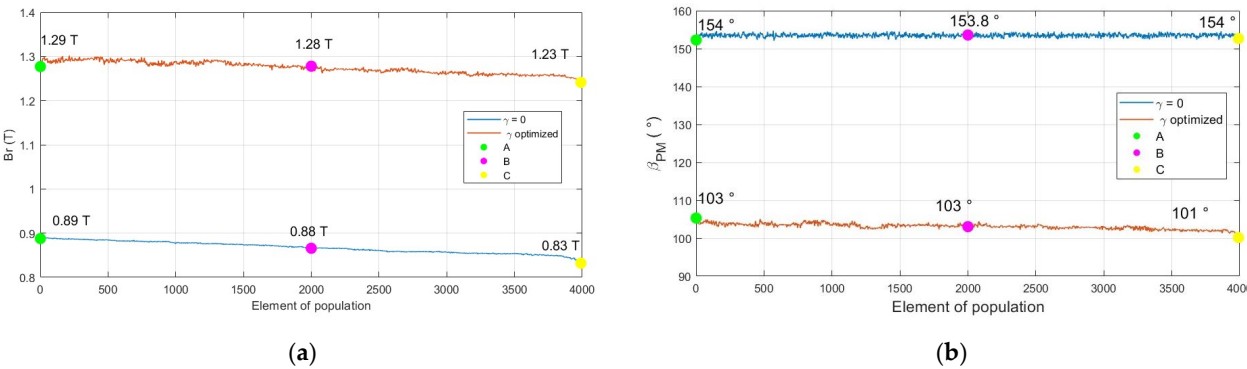

**Figure 7.** The evolution, for each element of the population, of the remanent flux density (**a**) and the electrical magnet pole arc (**b**).

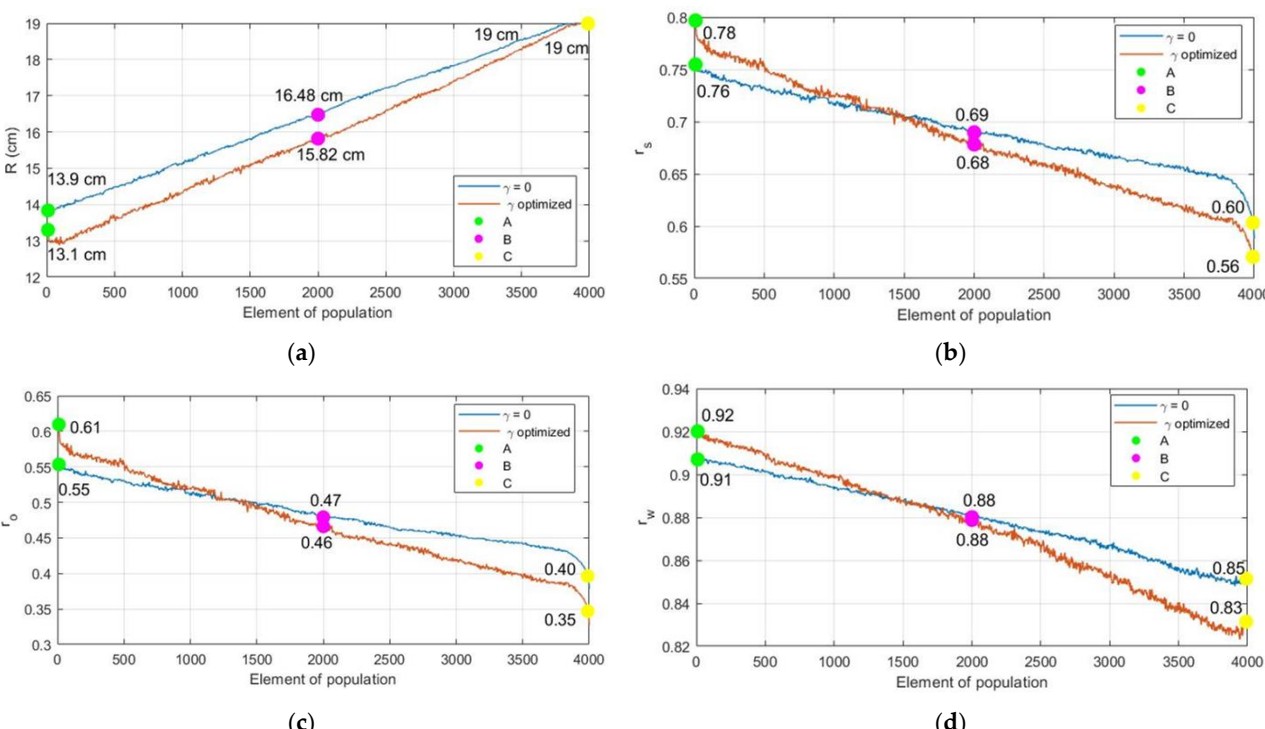

**Figure 8.** The evolution, for each element of population, of the stator's outer radius (**a**), the stator's inner reduced radius (**b**), the rotor's inner reduced radius (**c**), and the winding's outer reduced radius (**d**).

It can be seen that the remanent flux density of the magnets and their polar arcs are relatively constant along the front for both cases, $\gamma = 0$ and $\gamma$ optimized, showing that these parameters are not very sensitive to the optimization criterion (see Figure 7).

On the contrary, the geometric parameters are highly sensitive to each optimization criterion (mass or loss). The external radius, $R$, and volume decrease as expected with the minimization of the mass. Inversely, the pole pair number, $p$, and the reduced radii ($r_s$ and $r_w$) increase, resulting in a minimization of the yoke thickness until the limit is reached (see Figure 8).

The optimum results for the mass criterion (left end of the Pareto front) and the loss criterion (right end of the Pareto front) are shown in Tables 3 and 4.

**Table 3.** Optimum design parameters.

| Optimal Parameters | Mass Minimized | | Loss Minimized | | Combined Objectives | |
|---|---|---|---|---|---|---|
| | $\gamma=0$ | $\gamma_{opt}$ | $\gamma=0$ | $\gamma_{opt}$ | $\gamma=0$ | $\gamma_{opt}$ |
| $p$ | 4 | 5 | 2 | 2 | 3 | 3 |
| $R$ (cm) | 13.9 | 13.1 | 19 | 19 | 16.5 | 15.8 |
| $r_s$ | 0.76 | 0.78 | 0.6 | 0.56 | 0.69 | 0.68 |
| $r_w$ | 0.91 | 0.92 | 0.85 | 0.83 | 0.88 | 0.88 |
| $r_o$ | 0.55 | 0.61 | 0.4 | 0.35 | 0.47 | 0.46 |
| $\beta_{PM}$ (deg) | 154 | 103 | 154 | 101 | 154 | 103 |
| $W_{mag}$ (mm) | 15 | 15 | 15 | 15 | 15 | 15 |
| $B_r$ (T) | 0.89 | 1.29 | 0.83 | 1.23 | 0.88 | 1.28 |
| $\gamma$ | 0 | 0.2 | 0 | 0.2 | 0 | 0.21 |

**Table 4.** Optimum machine performance.

| | Mass Minimized | | Loss Minimized | | Combined Objectives | |
|---|---|---|---|---|---|---|
| | $\gamma = 0$ | $\gamma_{opt}$ | $\gamma = 0$ | $\gamma_{opt}$ | $\gamma = 0$ | $\gamma_{opt}$ |
| Mass (kg) | 24.9 | 20 | 66 | 67 | 40 | 34 |
| Total losses (kW) | 1.2 | 1.12 | 0.49 | 0.39 | 0.74 | 0.66 |
| Copper losses (kW) | 0.79 | 0.74 | 0.28 | 0.22 | 0.49 | 0.44 |
| Iron losses (kW) | 0.41 | 0.38 | 0.21 | 0.17 | 0.25 | 0.22 |
| $B_{sym}$ (T) | 1.6 | 1.6 | 1.34 | 1.36 | 1.5 | 1.51 |
| $B_{stm}$ (T) | 1.6 | 1.6 | 1.6 | 1.6 | 1.6 | 1.6 |
| $I_{RMS}$ (A) | 9.87 | 11.45 | 9.23 | 11.68 | 10.2 | 11.32 |
| $I_{1s}$ (A) | 9.87 | 11.21 | 9.23 | 11.32 | 10.2 | 11.1 |
| $I_{3s}$ (A) | 0 | 2.25 | 0 | 2.89 | 0 | 2.22 |
| $n_s$ | 102 | 68 | 198 | 166 | 130 | 107 |

We can observe, considering only the mass criterion (point A), that the optimal machine exploiting the third harmonic has better results than the machine exploiting only the fundamental harmonic, as its mass is 20% lower and the losses are 6.6% lower. It is observed that the maximum temperature is located at the end-windings and changes along the Pareto front, as illustrated in Figure 5a. As for the optimization considering the losses criterion alone (point C), the machine exploiting the third harmonic has once again better results concerning the losses, as they are around 21% lower compared to the machine exploiting only the fundamental harmonic, while for the masses, they are approximately the same for both machines. When considering both optimization criteria, it is observed that for the machine exploiting the third harmonic, the mass and losses are both lower (15% and 11%, respectively).

The torque density improvement, considering the given constraints, results from a better use of iron. The addition of a third harmonic in the stator induction can be achieved without a significant increase in iron flux densities. Among the few references dealing with the sizing of five-phase machines with the injection of a third harmonic, the ratio $\gamma$

is presented as the key parameter. Our results show that its optimum value is around 0.2, with limited sensitivity to the model. Whether we are considering the accuracy of the loss model (variation in the value of $k_{ad}$, for example) or the state of the constraints (maximum temperature reached or not), the optimum always oscillates around 0.2, with variations of just a few percent.

## 4. FE Validation

In order to validate the results of the optimization, a 2D finite element analysis is presented. For this part, we used the example of the machines designed according to the mass criterion during the optimization. Using the FEMM 4.2 software, the five-phase machine using only the fundamental-harmonic current and the machine with the third-harmonic current are both analyzed and validated. The magnitude of the flux densities and electromagnetic power are calculated and compared with the analytical results, as can be seen in Table 5. Figure 9 shows the flux lines and flux densities of one pole of the two machines using the parameters in Table 3. The main magnetic characteristics calculated with the FEM software are given in Table 5 and compared with those obtained by analytical calculation. The observed deviations remain below 10%, validating the proposed analytical model. The value of torque ripple [33–35], which is not calculated by the analytical model, is also checked. For the optimum geometries selected, this ripple is low.

**Table 5.** Comparison between the analytical model and FEM software.

| Cases | Quantity | Analytical Model | FE | Variation |
|:---:|:---:|:---:|:---:|:---:|
| | $C_{em}$ $(Nm)$ | 238 | 234 | 1.7% |
| $\gamma = 0$ | $B_{sym}$ $(T)$ | 1.6 | 1.53 | 4.4% |
| | $B_{stm}$ $(T)$ | 1.6 | 1.54 | 3.7% |
| | Torque ripple (%) | - | 3.14 | - |
| | $C_{em}$ $(Nm)$ | 238 | 230 | 3.3% |
| $\gamma_{opt}$ | $B_{sym}$ $(T)$ | 1.6 | 1.57 | 1.2% |
| | $B_{stm}$ $(T)$ | 1.6 | 1.65 | 3.1% |
| | Torque ripple (%) | - | 3.2 | - |

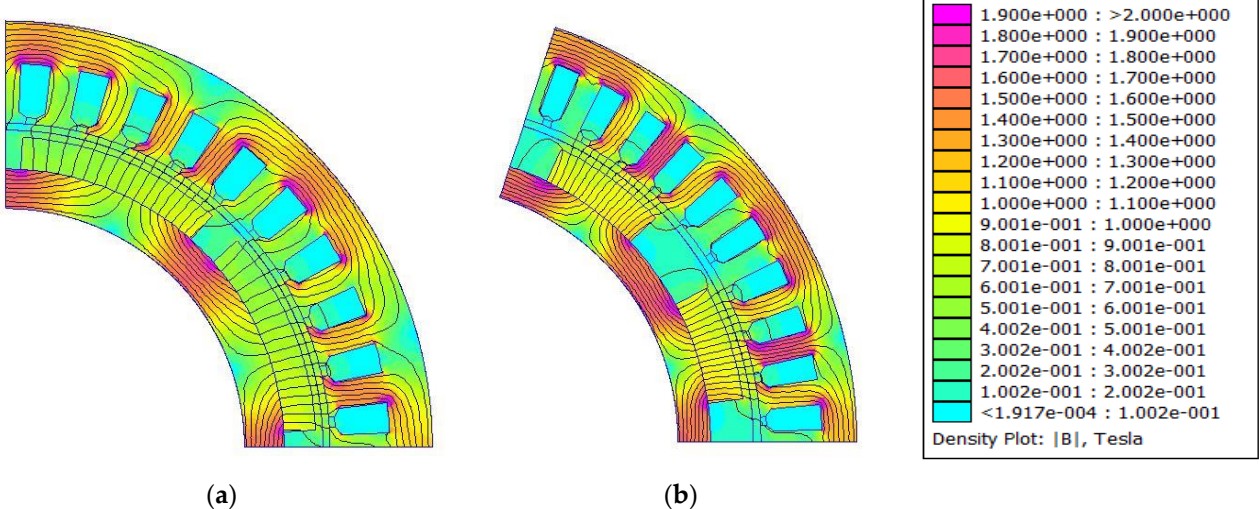

| | |
|:---:|:---:|
| (**a**) | (**b**) |

**Figure 9.** Flux lines and flux densities of the optimal machines: (**a**) optimal machine (mass minimized) with $\gamma = 0$; (**b**) optimal machine (mass minimized) with optimized $\gamma$.

## 5. Conclusions

In this paper, a study of the five-phase PMSM exploiting the third harmonic was carried out. A multiphysics analytical model of this machine was made, and an optimization problem respecting magnetic, thermal, electrical, and mechanical constraints was presented.

The study shows that the magnet properties (remanent flux density and opening arc) are not very sensitive to the chosen optimization criteria, unlike geometric parameters. As expected, the minimization of mass is achieved by increasing the number of pole pairs and minimizing the thickness of the yokes, whatever the optimization, with or without harmonic 3. The study clearly demonstrated the benefits of using a secondary machine. Considering the mass minimization only, the specific torque is increased by around 20%. Although the optimum rate of injection of the third-harmonic current has a relatively robust value (20%), it was also observed that this ratio is independent of temperature and saturation levels. The study therefore shows that it is necessary to take the third harmonic into account right from the design process. A machine sized on the mass criterion and for the fundamental harmonic only leads to a different design (a different number of pole pairs, etc.) and would not be able to operate with an additional third harmonic in steady state. Based on this work, the laboratory will present, in a future paper, the design optimization of polyphase machines considering their working cycle, taking into account a power converter. In particular, the aim will be to optimize the size of the machines, considering the control strategy for harmonics 1 and 3, depending on the operating point, with the objective of minimizing the energy lost during the cycle.

**Author Contributions:** Conceptualization, M.O., N.B., M.-F.B. and D.Z.; methodology, M.O. and N.B.; software, M.O. and N.B.; validation, M.O., N.B., M.-F.B. and D.Z.; formal analysis, M.O., N.B., M.-F.B. and D.Z.; investigation, M.O., N.B., M.-F.B. and D.Z.; resources, M.O., N.B., M.-F.B. and D.Z.; data curation, M.O., N.B., M.-F.B. and D.Z.; writing—original draft preparation, M.O. and N.B.; writing—review and editing, M.O., N.B.; visualization, M.O., N.B., M.-F.B. and D.Z.; supervision, N.B. and M.-F.B.; project administration, N.B. and M.-F.B. All authors have read and agreed to the published version of the manuscript.

**Funding:** This research received no external funding.

**Data Availability Statement:** Not applicable.

**Conflicts of Interest:** The authors declare no conflicts of interest.

## Abbreviations

| | |
|---|---|
| $A$ | area of the cylinder [m$^2$] |
| $B_r$ | remanent induction [T] |
| $B_{agm}$ | magnitude of the resultant airgap flux density [T] |
| $B_{fm}$ | airgap magnitude of the magnet flux density [T] |
| $B_{hs}$ | harmonic h airgap flux density [T] |
| $B_{rym}$ | magnitude of the flux density in the totor yoke [T] |
| $B_{sat}$ | saturation induction [T] |
| $B_{sm}$ | airgap magnitude creatd by the stator [T] |
| $B_{stm}$ | magnitude of the flux density in the stator teeth [T] |
| $B_{sym}$ | magnitude of the flux density in the stator yoke [T] |
| $\beta_{PM}$ | electrical magnet pole arc [deg] |
| $E_{hsk}$ | electromotive force of a phase k for an harmonic of rank h [V] |
| $E_{\alpha p}$ | electromotive force projected on the $\alpha_p$ axis [V] |
| $E_{\alpha s}$ | electromotive force projected on the $\alpha_s$ axis [V] |
| $E_{\beta p}$ | electromotive force projected on the $\beta_p$ axis [V] |
| $E_{\beta s}$ | electromotive force projected on the $\beta_s$ axis [V] |
| $f$ | frequency [Hz] |
| $Fmm_h$ | distribution of the harmonic electro-magnetomotive force |
| $h$ | harmonic rank |
| $h_x$ | convection heat coefficient [W/m$^2$K] |
| $I_{RMS}$ | RMS stator current [A] |
| $I_{sh}$ | current of harmonic h [A] |
| $k_{ad}$ | additional iron loss coefficient |

| | |
|---|---|
| $k_t$ | tooth-opening-to--slot-pitch ratio |
| $k_f$ | slot fill factor |
| $k_L$ | coefficient for correcting the active length |
| $K_H$ | hysteresis specific loss coefficient |
| $K_F$ | eddy currents specific loss coefficient |
| $L$ | active length [m] |
| $L_{ew}$ | end-winding length [m] |
| $L_0$ | self-inductance [H] |
| $L_p$ | cyclic inductance of the main machine [H] |
| $L_s$ | cyclic inductance of the secondary machine [H] |
| $M_c$ | mass of the copper [kg] |
| $M_{PM}$ | mass of permanent magnet [kg] |
| $M_{ry}$ | mass of the rotor yoke [kg] |
| $M_{st}$ | mass of the stator teeth [kg] |
| $M_{sy}$ | mass of the stator yoke [kg] |
| $n_s$ | number of turns per phase per pole |
| $p$ | number of pole pairs |
| $P_c$ | copper losses [W] |
| $P_{ew}$ | end-winding losses [W] |
| $P_{mgt}$ | iron losses in the stator teeth [W] |
| $P_{mgy}$ | iron losses in the stator yoke [W] |
| $P_{tot}$ | total losses of the machine [W] |
| $\mathcal{P}$ | surfacic permeance $\left[\text{N/mA}^2\right]$ |
| $q$ | number of phases |
| $R$ | outer stator radius [m] |
| $R_c$ | copper resistance [$\Omega$] |
| $R_{ew}$ | end-winding resistance [$\Omega$] |
| $\mathcal{R}_{rx}$ | radial thermal resistance [m²K/W] |
| $\mathcal{R}_t$ | orthoradial thermal resistance [m²K/W] |
| $\mathcal{R}_z$ | axial thermal resistance [m²K/W] |
| $\mathcal{R}_{cvx}$ | convection thermal resistance [m²K/W] |
| $r_s$ | stator inner reduced radius |
| $r_o$ | rotor inner reduced radius |
| $r_w$ | winding outer reduced radius |
| $r_{ext}$ | external reduced radius |
| $r_{int}$ | internal reduced radius |
| $S$ | rectangular cross-section [m²] |
| $S_s$ | slot cross-section [m²] |
| $th_{lam}$ | thickness of the lamination [m] |
| $U_{dc}$ | DC bus voltage [V] |
| $V_{limit}$ | voltage limit [V] |
| $W_{ag}$ | airgap thickness [m] |
| $W_{PM}$ | permanent magnet width [m] |
| $W_{mag}$ | magnetic airgap [m] |
| $W_{min}$ | minimum width of the yoke [m] |
| $W_{sy}$ | width of the satator yoke [m] |
| $W_{ry}$ | width of the rotor yoke[m] |
| $Z_s$ | number of slots |
| $\delta$ | skin depth [m] |
| $\rho_c$ | copper density $\left[\text{kg/m}^3\right]$ |
| $\rho_{Iron}$ | steel density $\left[\text{kg/m}^3\right]$ |
| $\rho_{PM}$ | permanent magnet density $\left[\text{kg/m}^3\right]$ |
| $\mu_0$ | vacuum permeability [N/A²] |
| $\mu_\mathbf{r}$ | material permeability |
| $\sigma_x$ | conductivity of the material [S/m] |
| $\psi_h$ | phase angle of harmonic h [deg] |
| $\Omega$ | machine mechanical angular velocity [rad/s] |
| $\lambda_x$ | material conductivity [S/m] |

| | |
|---|---|
| $\theta_\infty$ | ambient temperature [°C] |
| $\theta_{max}$ | maximal permissible temperature [°C] |
| $\tau$ | cylinder angle [deg] |
| $\gamma$ | ration between the third-harmonic current and the first-harmonic current |

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
