# Peer review of "Design and Optimization of a Five-Phase Permanent Magnet Synchronous Machine Exploiting the Fundamental and Third Harmonic"

_machines, doi:10.3390/machines12020117_

Round 1

Reviewer 1 Report

Comments and Suggestions for Authors

The authors in this paper are concerend with with the design of five-phase permanent magnet synchronous machines exploiting the third harmonic for torque generation. Through the optimization of the stator size and rotor structure, the objective functions related to the mass and electric losses can be optimized for a targeted electromagnetic power and a given volume. A 1-D analytical magnetic model. Please, consider the following issues that support you to improve the paper quality:

1. The abstract should be extedned to show the paper contribution with effective numerical contribution compared to previous methods. In other words, how  the optimal design change the toruqe and motor volume. 

2. The introduction section is very limited as several methods are exit in the literature to design such motor. Avoid compact and lumped references citation as [1-3], [4-8] and [11-13], [25-28] ....etc . Add the critical review of each reference to show the research gap. . Also, add the contribution in bullets followed by the paper orginization at the end of introdution section. 

3. Please, check typo errors and add the missed references: as in ( 1-Danalytical), As already demonstrated [20] [Ref], .etc

4. In Figures 1 and 2, add the defination of all used variables. You can collect all varaibles, parameters at the begining of introduction section. 

5. Extend your description of the concept of the three fictitious machines in section. 

6. For the flux densities in Eq. 1, add a reference with definition of the used varaibles after the equation or collected before introduction section not at the end of the paper as presented.  The same issue in all equations.

7. By the end of section 2.2 define the objectives you deduced. 

8. It is good to classfiy the constraints mechnical, electrical, thermal..etc. Add the related references with more discusssion. 

9. For equations, (22)-(25), V1s and V3s are the same as V1 and V3 or different? 

10. The thermal constraints are hard to follow and what are the types of these constraints? 

11. In section 3, provide the general mathermtical model based on the mentioned design basics of harmoinc and volume. 

12. How do you choose the upper bounds in equation (46)?

13. In Fig. 4, 15 phases or 5 phases you consider.

14. Extend the simulation results for indvidual or combined objectivees. 

15. The selection criteria of the optimization method and the success indinces must be discussed. 

16. Summerise your findings in the conclusion section as well as the future trend. 

17. Update your references with English words as: janv, Fevr, Julli, .....etc

Comments on the Quality of English Language

Moderate 

Reviewer 2 Report

Comments and Suggestions for Authors

All in one, the work seems quite interesting because focuses on the design of a 5-phase electrical machine, favouring the generation of extra torque using 3rd harmonic. However, the introduction section must be improve to state the real interest of the proposal. There are much modern references on the state of the art of multiphase machines that the one cited ([1] is from 2008...), most of the conference papers should be substitute by journal papers (I am sure you can find them), and all the references must be revised to cite more recent works. Moreover, the sentence "But few publications focus on the design optimization of the machine based on the use of the 3rd harmonic to increase power-to-weight ratio or efficiency." must be justified with references, and explain the research work of these references to justify the real interest of the work. Otherwise, the reviewers cannot appreciate the novelty.

Round 2

Reviewer 1 Report

Comments and Suggestions for Authors

The paper is improved but three issues are needed to be extended: 

11. In section 3, provide the general mathematical model based on the mentioned design basics of harmonic and volume. The description of equations 49-53 needs more attention. how do you optimize both objectives in 49? 

14. Extend the simulation results for individual or combined objectives.

15. The selection criteria of the optimization method and the success indices must be discussed.

16. Summerise your findings in the conclusion section as well as the future trend

Comments on the Quality of English Language

Moderate

Reviewer 2 Report

Comments and Suggestions for Authors

Thanks for your work. Please find attached some revisions about multiphase drives that, in my humble opinion, can improve the bibliography section (they all deal with 5-phase electrical drives and are recent reviews in IEEE and IET journals to attract future readers):

1.- Predictive current control in electrical drives: an illustrated review with case examples using a five-phase induction motor drive with distributed windings. M. Bermúdez et al. IET Electric Power Applications, Vol. 14, Iss. 8, pp. 1291-1310, 2020.

2.- Recent Advances in the Design, Modeling and Control of Multiphase Machines – Part 1. F. Barrero, M.J. Durán. IEEE Transactions on Industrial Electronics, Vol. 61, No. 1, pp. 449-458, 2016.

3.- Recent Advances in the Design, Modeling and Control of Multiphase Machines – Part 2M.J. Durán, F. Barrero. IEEE Transactions on Industrial Electronics, Vol. 61, No. 1, pp. 459-468, 2016. 

Round 3

Reviewer 1 Report

Comments and Suggestions for Authors

The paper is improved. 

Comments on the Quality of English Language

Moderate